# Multi-Process-Based Maximum Entropy Bootstrapping Estimator: Application for Net Foreign Direct Investment in ASEAN

**Arisara Romyen [1,2,*], Chukiat Chaiboonsri [1], Satawat Wannapan [1] and Songsak Sriboonchitta [1,3]**

1   Faculty of Economics, Chiang Mai University, Chiang Mai 50200, Thailand
2   Faculty of Economics, Prince of Songkla University, Songkhla 90110, Thailand
3   Puey Ungphakorn Center of Excellence in Econometrics, Faculty of Economics, Chiang Mai 50200, Thailand
*   Correspondence: arisara.r@psu.ac.th

**Abstract:** Due to a broad consensus in the engaging of global economic integrations, host countries encounter a number of challenges, especially in international capital mobility. Foreign direct investment (FDI) becomes a pillar for economic development. This study explores which Association of Southeast Asian Nations (ASEAN)-6 countries are good representatives to inform the directions of FDI. For computational modelling, the AR-GARCH model was created using the maximum entropy bootstrap estimation. Nonparametric techniques consisting of the maximum entropy bootstrap method and cross-entropy algorithm were applied. The results show that Indonesia has the nearest cross-entropy (CE) value compared to the whole entropy value, followed by Thailand and Singapore. Furthermore, it is consistent with the first- and second-order stochastic dominance analyses. Additionally, the structural dependence of capital movements is displayed to deeply investigate the capital flow relation among the countries. Consequently, the performances of FDI in Indonesia, Thailand, and Singapore can significantly convey the scenario of FDI across ASEAN.

**Keywords:** nonparametric methodology; maximum entropy bootstrap; cross-entropy analysis; stochastic dominance analysis; C-vine copula

**JEL Classification:** F43; C53

## 1. Introduction

Current time-series inferences heavily rely on the stationary process assumption that statistical properties are supposed to steady over time. In fact, a time series is often nonstationary, which can appear as seasonality, trends, random walks, or other evolutionary incidences. Such nonstationary series are by definition unpredictable and cannot be modelled. As a result, this violates the stationarity assumptions in the process of time-series analysis, and it may result in spurious and unreliable statistical inferences (Khinchine 1934; Kolmogorov 1931; Vinod 2006). In practice, a series tends to be constant in a short period and nonstationary in a longer duration. Additionally, an observed time series $\{x_t\}$ on random variable $X$ can perform as a stochastic process corresponding to the certain period of time ($t$), such as days, months, years, etc. It is important to take into account the presence of stationary and nonstationary series. Model misspecification is a regular problem in statistical data analysis in several methods. Such models offer biased coefficients and error terms, and these show invalid parameter estimations. Moreover, in the misspecified models dealing with the asymptotic theory, inferences employing usual statistics lead to spurious regressions (Phillips 1986; Shin and Sarkar 1997).

An updated technique undertaken in the nonparametric bootstrap methodology is the maximum entropy bootstrap (MEboot), proposed by Vinod (2006). The MEboot is a powerful tool for highly

dependent nonstationary time series and it can overcome unnecessary distributional assumptions of stationary. In addition, creating the simulations beyond the historical time series can guarantee what the relevant information means and how the future may unfold. The MEboot algorithm can intensively generate random samples called ensemble $\Omega$ based upon the empirical cumulative distribution function (CDF), which ensures it as an underlying process in cases of nonstationary or regime-switching in the time series (Vinod 2013; Vinod and Lopez-de-Lacalle 2009). This method also satisfies the ergodic theorem and the central limit theorem (Chaiboonsri and Chaitip 2013; Srivastav and Simonovic 2014).

In a broad swing of the global economy, the Association of Southeast Asian Nations (ASEAN) has been shifting towards trade liberalization and international capital mobility. Foreign direct investment (FDI) has been expanding and is greatly facilitated by agreement among trade partner countries, especially in advanced ASEAN economies consisting of Singapore, Thailand, Malaysia, Indonesia, Vietnam, and the Philippines that perform as the top six counties attracting FDI. Despite ASEAN being one of the major destinations of FDI and playing a crucial role in promoting economic growth in the region, each ASEAN member country should achieve its true potential, raising a question of who dominates FDI in ASEAN economy. Having said that, the global competition in trade liberalization comes up with high challenges to the host countries, so it is necessary for ASEAN member states to know their true competitive positions for investment in order to prepare more FDI attractive activities, and the competitive positioning is the ultimate concern for long-term performance of FDI.

For that reason, simulation-based econometrics such as the entropy-based inferential models including maximum entropy (ME) method, cross-entropy (CE) algorithm, and the MEboot approach are particularly applicable to reach the best experimental model fitting. The objectives of this research are to computationally seek the efficient estimator by applying empirical evidence to determine the predominant nation of FDI attractiveness and to deeply clarify the structural dependence of capital flows among ASEAN countries. The rest of this article is organized as follows. We briefly summarize the maximum entropy bootstrapping estimator, maximum entropy and cross-entropy principles, stochastic dominance criteria, and C-vine copula model. In Section 3, data descriptive, empirical applications of the proposed methods are presented, and Section 4 concludes.

## 2. Methodology

The procedure of this research can be broadly divided into five parts, as detailed in the following sections.

### 2.1. AR-GARCH Model

Recently, the world economy has confronted economic crises due to the globalized economic environment. Likewise, the uncertainty on FDI flows has commonly appeared in host countries. Such volatile FDI may affect those macroeconomies; underlying the interaction behind volatile FDI is essential. To capture suitably the unobservable process, the volatile series of FDI is generated via a class of general autoregressive processes under white noises participating conditional heteroscedastic variances, which is the GARCH-type modelling. The growth rate of FDI, namely $\{x_t\}$, is called an autoregressive process of order $k$ with a GARCH noise for order $p,q$ for $t = 0, \pm 1, \pm 2, \ldots$ Therefore, the AR($k$)-GARCH ($p,q$) processes proposed by Bollerslev (1986) are defined by the following three equations:

$$x_t = \sum_{i=1}^{k} \alpha_i x_{t-i} + \varepsilon_t \tag{1}$$

$$\varepsilon_t = \sigma_t v_t \tag{2}$$

$$\sigma_t^2 = \varphi_0 + \varphi_1 \varepsilon_{t-1}^2 + \ldots + \varphi_p \varepsilon_{t-p}^2 + \beta_1 \sigma_{t-1}^2 + \ldots + \beta_q \sigma_{t-q}^2. \tag{3}$$

where $x_t$ is the growth rate of FDI for each nation, $\alpha_i$ and $k$ in Equation (1) represent the parameters and the order of AR, $v_t$ in Equation (2) performs as a white noise (iid(0,1)), and $\sigma_t$ satisfies Equation

(3). $\varphi_i$ and $\beta_i$ in Equation (3) refer to the parameters of GARCH ($p,q$). Then, Var $(\varepsilon_t|\varepsilon_{t-1},\ \varepsilon_{t-2},\dots) = \sigma_t^2$, $E(\varepsilon_t) = 0$, $Cov(\varepsilon_t, \varepsilon_s) = 0$, and $t \neq s$. Hence, we can employ these estimations to obtain the generalized residuals and further calculate the CDF to carry out the maximum entropy.

*2.2. Maximum Entropy Bootstrapping Estimator (MEboot)*

Vinod (2006), and further studied by Vinod and Lopez-de-Lacalle (2009), invented the MEboot approach, which is a technique for bootstrapping time series to avoid unnecessary distributional assumptions like unit root and structural change relating to shape-destroying conversions and complicated asymptotic to attain stationarity. The maximum entropy bootstrapping estimator (MEboot) can use to treat the likelihood and obtain unknown parameters. Moreover, the MEboot approach efficiently offers a certain parameter rather than employing the maximum likelihood estimator (MLE) (Wannapan Satawat 2018). The MEboot evokes a maximum entropy density $f(x)$ respective to certain mass-and-mean-preserving constraints. Let $f(x)$ be the density of $x_t$, thus the maximizing the Shannon information is defined by:

$$H = E(-log f(x)). \tag{4}$$

According to Vinod (2006), an intensive formation of a plausible ensemble $\Omega$ generated from a density was accomplished and satisfied the maximum entropy (ME) principle. The MEboot algorithm employs quantiles, which are routed through the $\Omega$ from the inverse evident CDF of the ME density, denoting $\{x_{j,t}\}$, for $j = 1,\dots J,\ J+1,\dots$. The entire mean of all $\{x_{j,t}\}$ is definitely equivalent to the mean of $\{x_{j,t}\}$ ex post without a doubt of asymptotic behaviors over time points ($T$). The constructed replicates satisfy the ergodic theorem and the central limit theorem and ensure to preserve the original properties of a time series, such as shapes, autocorrelation, and partial autocorrelation functions. Chaitip and Chaiboonsri (2013) concisely summarized Vinod's seven-step to MEboot algorithm to generate a random realization of $x_t$.

*2.3. Maximum Entropy (ME) Principle*

The principle of ME is to extract meaningful constraints that predicate the observed signals originated by the system. Following the concept of Jaynes (1963) since 1957, if the probability distribution function (PDF) of a given parameter $X$, being continuous distribution, is unknown and some parts of the parameter distribution are known, we can adopt the ME algorithm as demonstrated by Muoz-Cobo et al. (2017) to obtain the parameter distribution. In the case of $X$ taking a compact aspect being $[a; b]$, with $b > a$, the Shannon information entropy is expressed:

$$H = -\int_a^b f_X(x) \log(f_X(x)) dx = -\int_a^b \log(f_X(x)) dF_X, \tag{5}$$

where $H$ denotes the information entropy defined by Shannon (1948). $f_X(x)$ is the PDF of the random parameter $X$, and $F_X(x)$ is the CDF. Considering the PDF in general form in Equation (6), the $g_i(x)$ functions assign the different moments of the distribution function of the parameter $X$, where the number $i$ is taken from 1 to $n$ as follows:

$$\int_a^b g_i(x) f_X(x) dx = u(g_i) = u_i, \ i = 0,\ 1,\ 2,\ \dots, n. \tag{6}$$

The ME can be solved to achieve the PDF definition $f_X(x)$ that maximizes the information entropy. Taking the Lagrange multiplier method of Equation (6) gives:

$$J[f_X] = -\int_a^b f_X(x) \log(f_X(x)) dx + \sum_{i=0}^n \lambda_i \left[ \int_a^b dx g_i(x) f_X(x) - u_i \right], \tag{7}$$

The first variation of the functional $J[f_X(x)]$ is $\delta J[f_X(x)]$ and the interference $\delta f_X(x)$ equals to zero, and we receive the PDF of the parameter as:

$$f_X(x) = exp\left\{-1 + \sum_{i=0}^{N} \lambda_i g_i(x)\right\}.$$  (8)

More generally, a nonlinear approach is required to obtain the values of $\lambda_i$, which are procured from the observed information on the distribution moments.

### 2.4. Cross-Entropy (CE) Analysis

A general problem in various fields of economic research is finding the expected value of a random quantity such as:

$$\alpha := E_f[\mathcal{N}(X)],$$  (9)

where $X = (X_1, \ldots, X_n) \in \mathbb{R}^n$ denotes a vector with the PDF $f(x)$, and $\mathcal{N}$ refers to an arbitrary real-valued function in $\mathbb{R}^n$. To estimate $\alpha$, the samples $X^1, \ldots, X^n$ are assumed to be independent and identically distributed (*i.i.d.*) from an approximately chosen PDF $g(\cdot)$, and the $\alpha$ is estimated by:

$$\hat{\alpha}_N(g) = \frac{1}{N} \sum_{j=1}^{N} \mathcal{N}(X^j) \frac{f(X^j)}{g(X^j)}.$$  (10)

The PDF $g(\cdot)$ has to dominate $\mathcal{N}(\cdot)$ $f(\cdot)$ in the absolutely continuous aspect. This quotes, *Supp* $[\mathcal{N}(\cdot)f(\cdot)] \subset Supp [g(\cdot)]$, where *Supp* stands for the support of the corresponding function. We seek to find the best parameter $\theta^*$. At the present, Rubinstein (1997) introduced the cross-entropy method to estimate the parameter $\theta^*$ such that g($\cdot$; θ∗) minimizes the Kullback–Leibler cross entropy corresponding to the zero-variance PDF $g^*$. In common, a certain stochastic optimization problem is necessary to solve for $\theta^*$ (Homem-de-Mello 2007).

### 2.5. Stochastic Dominance Analysis

Given $X_1$ and $X_2$ be the two interesting variables at any point in time and the observations, $X_{ki}, i = 1, \ldots, N; K = 1, 2$ are not necessarily *i.i.d.*, suppose that $U_1$ refers to the sequence of all Von Neumann–Morgenstern utility functions, $u$, such that $u' > O$ (increasing). $U_2$ stands for the class of all utility functions in $u_1$ within $u'' \ll O$ (strict concavity). Let $F_1(x)$ and $F_2(x)$ denote the CDF, respectively.

**Definition 1.** $X_1$ *first-order stochastic dominates* $X_2$*, denoted that* $X_{1 \geqslant FSD} X_2$*, if and only if:*

*(1)  $E[u(X_1)] \geq E[u(X_2)]$ for all $\mu \in U_1$ with strict inequality for some $\mu$; or*
*(2)  $F_1(x) \leq F_2(x)$ for all $x$ with strict inequality for some $x$.*

**Definition 2.** $X_1$ *second-order stochastic dominates* $X_2$*, denoted that* $X_{1 \geqslant SSD} X_2$*, if and only if either:*

*(1)  $E[u(X_1)] \geq E[u(X_2)]$ for all $\mu \in U_2$ with strict inequality for some $\mu$; or*
*(2)  $\int_{-\infty}^{x} F_1(t)dt_1 \leq \int_{-\infty}^{x} F_2(t)dt$ for all $x$ with strict inequality for some $x$.*

### 2.6. The C-Vine Copula Model

The fundamental theorem is based on the concept of Sklar (1959), and this can be shown in Equation (13) as follow:

$$F(x_1, x_2, \ldots, x_n) = C(F_1(x_1), F_2(x_2), \ldots, F_n(x_n)).$$  (11)

*F*: *n*-dimensional distribution with marginal $F_i$, $i = 1, 2, \ldots, n$;

$F_1, \ldots, F_n$: random vectors;

*C*: *n*-copula for all $X_1, X_2, \ldots, X_n$.

The function *C* is a distribution function that has uniform margins between zero and one, and it is labelled as the copula function. It binds the univariate margins $F_1$ and $F_2$ to produce bivariate distribution *F*.

The vine copula models are a graphical representation to specify pair copula constructions (PCCs). Basically, a principle for constructing multivariate copula generated from the product of bivariate pair copula was statistically explained as canonical (C-) vines. This contribution was a flexible model since bivariate copulas can accommodate complex structural dependences such as asymmetric dependences or strong joint-tail behaviors (Nikoloulopoulos et al. 2012; Charfeddine and Benlagha 2016). Consequently, the estimated patterns of relation among FDI flows in ASEAN are defined as $X = x_1, x_2, x_3, x_4, x_5, x_6$ with marginal distribution function $F_1, F_2, F_3, F_4, F_5, F_6$. The *n*-dimensional density (*n* = 6) corresponding to a C-vine copula is formulated as:

$$C_{12\ldots n}[F_1(x_1), F_2(x_2), \ldots, F_n(x_n)]$$
$$= \prod_{j=1}^{n-1} \prod_{k=1}^{n-j} c_{j,j+k|1,\ldots,j-1}[F(x_j|x_1,\ldots,x_{j-1}), F(x_{j+k}|x_1,\ldots,x_{j-1})], \tag{12}$$

where the C-vine comprises of five trees (*j* = 1, 2, ... , 5) and 15 edges. Each edge associates with a pair-copula. Then, the C-vine copula log-likelihood function is defined as:

$$L(x_1, \ldots, x_n; \theta) = \sum_{j=1}^{n-1} \sum_{k=1}^{n-j} \sum_{t=1}^{\tau} \log\big(c_{j,j+k|1,\ldots,j-1}[F(x_j|x_1,\ldots,x_{j-1}), F(x_{j+k}|x_1,\ldots,x_{j-1})]\big). \tag{13}$$

where $\theta$ denotes a set of the C-vine parameters and the time series contains $\tau$ independent observations. So, a C vine with six variables, five trees and 15 edges is displayed in Figure 1.

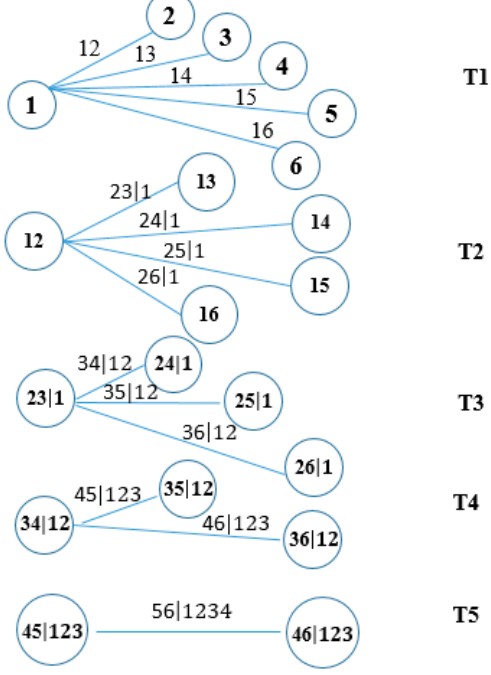

**Figure 1.** A canonical vine with six variables, five trees and 15 edges.

### 3. Data and Empirical Results

In this section, we present data description. Then, we show the application for the computational experiment for the past observations of a time series.

*3.1. Data Analysis*

To examine which country is the main indicator for FDI inflows in ASEAN, a series of FDI from the annual reports by the World Bank Development Indicators (WDI) database were collected during 1970 to 2017. For a brief insight of the underlying data, the plots of FDI series display a growing trend over full periods and are not normally distributed for all ASEAN-6 countries, as shown in Figure 2.

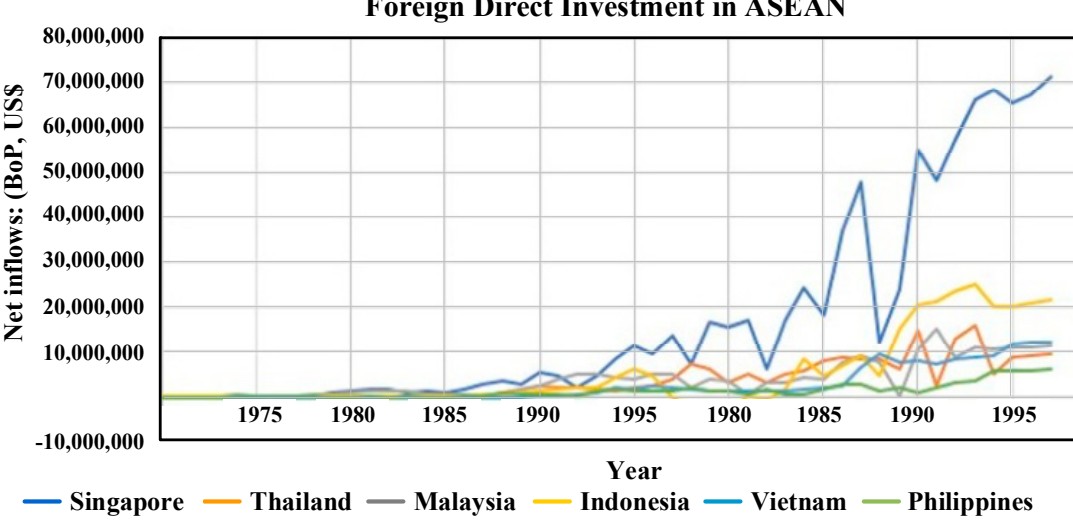

**Figure 2.** Plots of foreign direct investment (FDI); net inflows (BoP, current US$) in ASEAN-6 countries.

Then, those FDI series were transformed to be a growth rate, which typically shows the changed annualized rate of growth of the FDI in order to inform certain properties of parameterized distributions. Figure 3 demonstrates the plots of the FDI's growth rate. It can be clearly seen that there was low stability at the beginning of the FDI historical performance in Vietnam. Later, Vietnam established a transitional economy and opened the economy to the global market. As a result, it was extraordinary for Vietnam to enhance FDI from 1986 up to mid-1990 (Schaumburg-Mller 2002). Whereas, other nations have remained constant in FDI scales, excepting Malaysia that increased dramatically in 2010 due to its stimulating FDI policy.

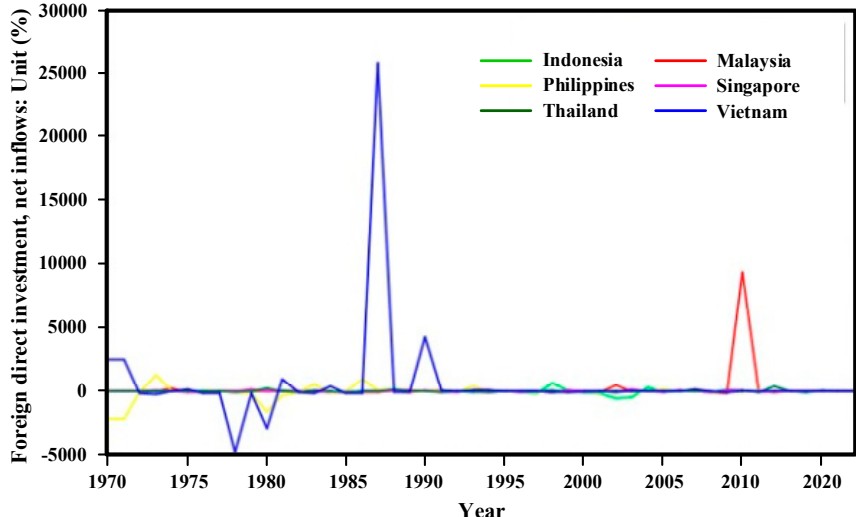

**Figure 3.** Plots of the FDI's growth rates.

Table 1 summarizes the descriptive statistics involving mean–medium and maximum–minimum FDI volumes, standard deviation, and normality properties. To simply determine whether the data set is a normal distribution, the skewness values of all nations were found, and they are greater than 1, indicating that the distribution is highly skewed. The kurtosis ranks for the whole are also leptokurtic. Moreover, the big Jarque–Bera test value and the tiny probability mean that the null hypothesis of normality distribution is rejected at the 5% significance level. Consequently, the FDI data series for all countries have no normal distribution. Even though Singapore is the largest destination of FDI inflows, they would not be able to utilize the mean value to perform as a higher position country in terms of averaging to dominate other countries in ASEAN.

**Table 1.** Descriptive data of the FDI.

|  | Singapore | Thailand | Malaysia | Indonesia | Vietnam | Philippines |
|---|---|---|---|---|---|---|
| Mean | $1.72 \times 10^{10}$ | $3.75 \times 10^{9}$ | $3.92 \times 10^{9}$ | $5.05 \times 10^{9}$ | $2.64 \times 10^{9}$ | $1.34 \times 10^{9}$ |
| Median | $5.87 \times 10^{9}$ | $2.09 \times 10^{9}$ | $2.76 \times 10^{9}$ | $6.29 \times 10^{8}$ | $1.11 \times 10^{9}$ | $6.76 \times 10^{8}$ |
| Maximum | $7.13 \times 10^{10}$ | $1.59 \times 10^{10}$ | $1.51 \times 10^{10}$ | $2.51 \times 10^{10}$ | $1.23 \times 10^{10}$ | $6.08 \times 10^{9}$ |
| Minimum | $9.30 \times 10^{7}$ | $5.53 \times 10^{7}$ | $9.40 \times 10^{7}$ | $-4.55 \times 10^{9}$ | $-8.90 \times 10^{5}$ | $-1.06 \times 10^{8}$ |
| Std.Dev. | $2.28 \times 10^{10}$ | $4.23 \times 10^{9}$ | $4.05 \times 10^{9}$ | $8.19 \times 10^{9}$ | $3.84 \times 10^{9}$ | $1.69 \times 10^{9}$ |
| Skewness | 1.332 | 1.159 | 1.021 | 1.334 | 1.367 | 1.671 |
| Kurtosis | 3.281 | 3.550 | 2.938 | 3.282 | 3.391 | 5.037 |
| Jarque–Bera | 14.347 | 11.363 | 8.346 | 14.399 | 15.262 | 30.655 |
| Probability | 0.001 | 0.0153 | 0.000 | 0.001 | 0.003 | 0.000 |
| Sum | $8.25 \times 10^{11}$ | $1.80 \times 10^{11}$ | $1.88 \times 10^{11}$ | $2.42 \times 10^{11}$ | $1.27 \times 10^{11}$ | $6.43 \times 10^{10}$ |
| Sum Sq.Dev. | $2.45 \times 10^{22}$ | $8.40 \times 10^{20}$ | $7.71 \times 10^{20}$ | $3.15 \times 10^{21}$ | $6.94 \times 10^{20}$ | $1.33 \times 10^{20}$ |

*3.2. Correlation Analysis*

In order to explore who dominates FDI direction in ASEAN, we initially began by determining if there is an association between two countries. Table 2 represents the correlation analyses in terms of parametric and nonparametric correlations. According to the simple Person's correlation, it indicates that Singapore has a mostly positive relationship to other countries considered. It should be noted that the Person's correlation analysis attempts to draw a linear association between two variables, but those variables might be inconsistent. If the relationship is not linear, then the interpretation is meaningless. To further verify that those certain variables are related to each other, we used the nonparametric correlations (Kendall's tau and Spearman's rho) to measure the strength and direction of association between the two involved variables. Unlike with the Person's correlation, Indonesia has a strong and

positive relationship to other countries at the 0.01 and 0.05 levels of significance of both Kendall's tau and Spearman's rho ranks.

**Table 2.** Person's correlation, Kendall's tau and Spearman's rank correlation.

| Correlations | Singapore | Thailand | Malaysia | Indonesia | Vietnam | Philippines |
|---|---|---|---|---|---|---|
| **Parametric** | | | | | | |
| Singapore | **1.000** | 0.081 | 0.248 | −0.016 | 0.117 | 0.063 |
| Thailand | **0.081** | 1.000 | 0.196 | 0.067 | −0.015 | −0.079 |
| Malaysia | **0.248** | 0.169 | 1.000 | −0.008 | −0.026 | 0.001 |
| Indonesia | **−0.016** | 0.067 | −0.008 | 1.000 | 0.037 | 0.020 |
| Vietnam | **0.117** | −0.015 | −0.026 | 0.037 | 1.000 | −0.015 |
| Philippines | **0.063** | −0.079 | 0.001 | 0.020 | −0.015 | 1.000 |
| **Nonparametric** | | | | | | |
| **Kendall's Tau** | | | | | | |
| Singapore | 1.000 | 0.096 | 0.092 | **0.268 \*\*** | −0.015 | −0.066 |
| Thailand | 0.096 | 1.000 | −0.021 | **0.089** | −0.022 | −0.034 |
| Malaysia | 0.092 | −0.021 | 1.000 | **0.116** | −0.047 | −0.073 |
| Indonesia | 0.268 \*\* | 0.089 | 0.116 | **1.000** | 0.067 | 0.036 |
| Vietnam | −0.015 | −0.022 | −0.047 | **0.067** | 1.000 | −0.090 |
| Philippines | −0.066 | −0.034 | −0.073 | **0.036** | −0.090 | 1.000 |
| **Spearman's rho** | | | | | | |
| Singapore | 1.000 | 0.152 | 0.117 | **0.340 \*** | −0.020 | −0.091 |
| Thailand | 0.152 | 1.000 | −0.031 | **0.122** | −0.022 | −0.013 |
| Malaysia | 0.117 | −0.031 | 1.000 | **0.111** | −0.067 | −0.101 |
| Indonesia | 0.340 \* | 0.122 | 0.111 | **1.000** | 0.088 | 0.050 |
| Vietnam | 1.000 | 0.096 | 0.092 | **0.268 \*\*** | 1.000 | −0.066 |
| Philippines | 0.096 | 1.000 | −0.021 | **0.089** | −0.015 | 1.000 |

\*\* Correlation is significant at the 0.01 level (2-tailed); \* Correlation is significant at the 0.05 level (2-tailed).

### 3.3. The Empirical Applications of the Entropy-Based Inferential Models

Afterwards, we constructed the AR-GARCH model. The simulated data sets were carried out using the MEboot estimation, which efficiently offers a precise parameter. The residuals were obtained and converted into the CDF terms, which execute to be a real-valued random variable. Then, we utilized those CDF sets to compute on the entropy formula. Furthermore, the cross-entropy approach was calculated to implicitly measure the minimum underlying set of events. The results of the entropy and the CE method are reported in Table 3. Empirically, Indonesia has apparently the nearest CE value (8.083) comparing to the overall Entropy value (6.491), followed by Thailand (8.245) and Singapore (8.504). Theoretically, the CE approach offers more accuracy to solve classification problems since it computes the actual probability of FDI series *(X)* for each time *t* associated with the realized entropy (*Q*) following a particular probability distribution within the entire observations. Like using a neural network to execute classification, this calculation statistically yields to the CE between *X* and *Q*, in which we can verify and evaluate the division of FDI. In short, we are cross-checking which country dominates FDI direction in ASEAN economy. Consequently, Indonesia, Thailand, and Singapore can be viewed as the key indicators of FDI inflows among ASEAN-6 host economies. It is consistent with the nonparametric correlation analysis above, implying that the flow directions of FDI in Indonesia, Thailand, and Singapore are more straightforward than elsewhere in ASEAN.

**Table 3.** Results of the general entropy and the cross-entropy.

| Country | CE Analysis Using MEboot | Overall Entropy | Ranking |
|---|---|---|---|
| Singapore | 8.505 | | 3 |
| Thailand | 8.245 | | 2 |
| Malaysia | 8.804 | 6.491 | 4 |
| **Indonesia** | **8.083** [a] | | **1** |
| Vietnam | 8.880 | | 5 |
| The Philippines | 8.983 | | 6 |

[a] the nearest cross-entropy (CE) value to the overall entropy value using MEboot.

Interestingly, Indonesia in recent years is one of the most popular prospective FDI host countries for FDI, and its FDI growth coincides with the global FDI growth. The FDI is playing a crucial role in the economic expansion, mainly driven by mining, chemical, pharmaceutical, transportation, and telecommunication industries. The Government has reformed many regulations and bureaucracies to create a good investment climate, and its global credit rating can also be graded. This influences the country's allure as an investment destination, providing its wealth of good strategies, political stability, and skilled workforce. Thailand is becoming more attractive as an FDI destination. The government has promoted several investment funds for technology and infrastructure to enhance in public–private partnerships. The target FDI sectors are in the areas of commerce, entrepreneurship, and innovation.

Even Singapore remains center stage as a regional hub for oversea trade and investment. It is relatively the largest share recipient of FDI in ASEAN. Though, those FDI inflows are mainly classified in the financial and insurance services and wholesale and retail trade sectors, while the manufacturing and real estate sectors play only a minor capacity. Consequently, this seems to be a weak link between recorded FDI flows and real economic activities in foreign-owned companies in Singapore (Sjöholm 2016). FDI performances undertaken by Singapore are likely untrustworthy. In addition, since Singapore is voluntarily open internationally, its economy indeed depends on the global economy. Hence, it is vulnerable with the world economy and its main trading partner's economic situations.

*3.4. Stochastic Dominance Analysis*

To deeply look inside more accurate results of the FDI analyses, we applied the stochastic dominance-based entropy approach between Indonesia and Singapore. Let *F* and *G* denote the CDF of FDI for Indonesia and Singapore, respectively. It is clearly seen that the distribution of FDI-based MEboot method in Indonesia dominates that of the relevant parts in Singapore's both first- and second-order stochastic dominance analyses, as depicted in Figure 4b,c.

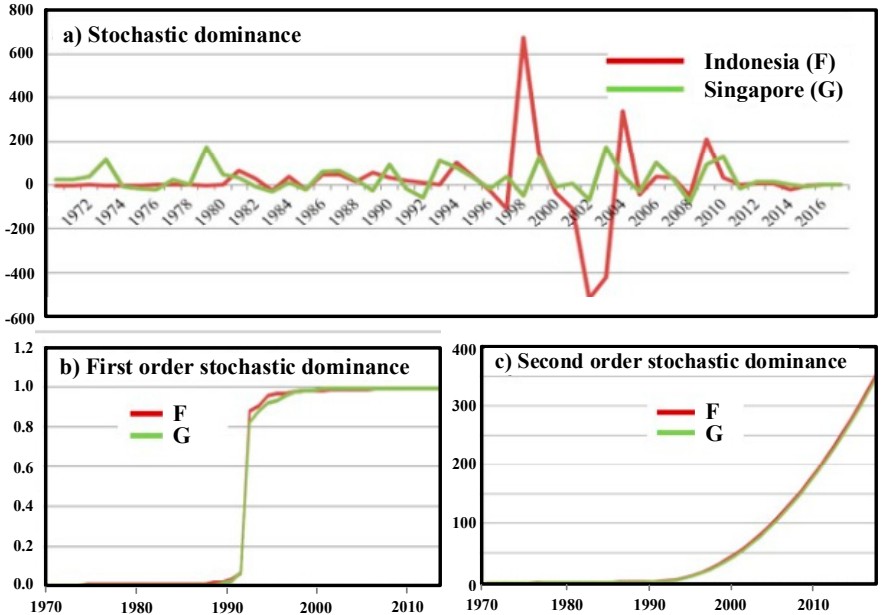

**Figure 4.** First- and second-order stochastic dominance analyses.

Since FDI occupies a special place in the link between economic progression and globalization, consequently, each host country should reach its actual relative position of FDI against competitors in order to enhance more competitiveness. This finding reveals that the directions of FDI in Indonesia (in particular), Thailand, and Singapore are the good representatives of all ASEAN-6 nations. In response, governments elsewhere should recognize that Indonesia, Thailand, and Singapore can display vital signs or leadership to boost up or slow down the FDI situation across ASEAN region. On the other hand, Malaysia, Vietnam, and the Philippines performed as followers and should improve their competitiveness by using government policies for attracting FDI, such as reforming business regulations, reforming and opening up policies, etc.

### 3.5. The Structural Dependence of Capital Flows in ASEAN by the C-Vine Copula Approach

Technically, the FDI in Indonesia is determined as a leadership, which is the predominant sign for capital movements among ASEAN-6 nations. The consequence of variables is evidently supported by the CE and the stochastic dominant analyses as mentioned above. In the first levels, we estimated the empirical dependence measure using pairwise maximum likelihood estimation. Based on the C-vine copula structure (six variables, five5 trees and 15 edges), the estimated C-vine pair-copula parameters are 0.19, 0.38, 0.07, 0.05, 0.01, 0.22, 0.04, 0.19, $-0.01$, 0.45, $-0.01$, $-0.01$, 0.15, $-0.03$, and 0.02, and the log likelihood is 27.02. Then, we employed all of those relevant variables, which maximize the sum of empirical dependencies using the spanning tree algorithm, to graphically draw the tree of the specified C-vine corresponding to the pair-copula parameters as edge labels. Considering the details of Figure 5, it is obvious that structural dependences dominated by Indonesian's net capital flows are positive and influence the FDI directions for others. To summarize, Indonesia shows moderately positive dependencies with Singapore, within a Gaussian copula ($\mathcal{N}$) with correlation $\rho = 0.5$, and Thailand ($\mathcal{N}$, 0.11), whereas it has relatively low effects on Malaysia ($\mathcal{N}$, 0.03), Vietnam ($\mathcal{N}$, 0.03), and the Philippines ($\mathcal{N}$, 0.01).

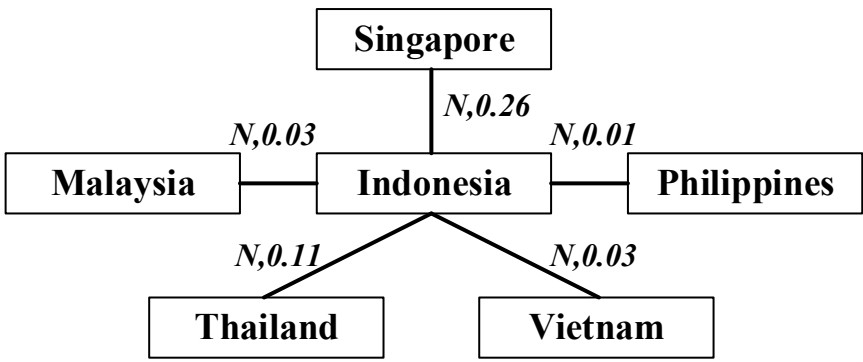

**Figure 5.** The C-vine copula analysis.

## 4. Conclusions

Since a time series regularly behaves as stochastic processes, the approaches using normal statistics may occur misspecified models. In the global economy, FDI has long been a key pillar in investment dynamic. Especially in ASEAN economy, FDI can enhance economic expansion at various levels of development. FDI flows to ASEAN have been rising since 2000, and the upward trend is reflective of attraction and confidence for oversea investment in the region. This causes ASEAN member states to rely more on the adoption of FDI. Then, this study sought to measure the competitive positions of each ASEAN nation. The findings would enhance the competitiveness of those ASEAN countries to understand, monitor and expedite FDI strategies in order to achieve effective aboard investment and attain competitive advantages across ASEAN economy. Consequently, we adopted the nonparametric methodology for multiprocessing analyses, including maximum entropy bootstrap method, cross-entropy algorithm, and the stochastic dominance analyses. For computational modelling, the AR-GARCH model was generated to obtain residual terms using the MEboot method. Furthermore, the C-vine copula model was applied to determine the structural dependence of FDI in ASEAN. The main findings can be drawn as follows:

- From basically testing and measuring the correlations between two variables, we found Indonesia has a more positive influence than other nations using the nonparametric correlations of Kendall's tau and Spearman's rho at the 0.01 and 0.05 levels of significance ranks.
- For advanced investigations on the classification tasks, the CE was preferred instead of the classification on parametric correlation, since it incorporates with entropy (information content) when handling all of the probabilities. It works with a very specific set of possible output values to evaluate the quality of the system network. Empirically, Indonesia performs the narrowest CE point compared to the overall Entropy point, followed by Thailand and Singapore, implying that Indonesia, Thailand, and Singapore can be identified as the main indicators for the FDI directions in the ASEAN.
- Being precisely supported by the first- and second-order stochastic dominance analyses, Indonesia is perceived as a leading indicator of FDI direction in ASEAN.
- Moreover, the structural dependence model called the C-vine copula strongly emphasized that the net capital flows in ASEAN rely on the capital movements in Indonesia. The positive dependences are obvious for the overall analysis.

This study has some policy implications. Therefore, the performances of FDI in Indonesia, Thailand, and Singapore can be evidently viewed as the key indicators for trends and developments of FDI in the ASEAN region. The Governments of Malaysia, Vietnam, and the Philippines should create more incentives toward FDI policies, such as competitive positioning, investment promotion, a degree of economic stability, etc. According to a positive correlation between FDI inflows, the enhancement to strengthen the ASEAN economic integration can bring benefits across the ASEAN economy.

**Author Contributions:** A.R. designed the main writer of the paper, collected data and analyzed the results. C.C. and S.W. constructed the model analysis and source coding for the R software. S.S. dedicated the overall research and revised the empirical findings and policy recommendations. All authors revised and approved the final manuscript.

**Funding:** This research received no external funding.

**Conflicts of Interest:** The authors declare no conflict of interest.

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
