# Peer review of "Multi-Process-Based Maximum Entropy Bootstrapping Estimator: Application for Net Foreign Direct Investment in ASEAN"

_economies, doi:10.3390/economies7030064_

Round 1

Reviewer 1 Report

I have read the submitted paper with interest. Though, I have identified several weaknesses:

there are grammar mistakes/deficiencies in a manuscript, e.g., see "Additionally, substantial applications to real time series data involve a combination of stationary and 25 non-stationary. Therefore, there frequently suffer the violation of stationary assumption. 

some formulas seem pasted probably from other pdf file (they seem like an effect of a snapshot tool use)

poor editing of the text, e.g., "One may be misspecified, the order of a non-stationary autoregression; One may regress"

the description of methods is too extensive, it seems like a lecture to students, the description should be shorter and refer to the previous empirical studies run with those methods, preferably in international economics context

the paper lacks discussion of results, there are no mentions about possible extensions and limitations of the paper

There are too many value judgments, e.g., "Each ASEAN member should know its true competitive position".

Author Response

a point-by-point response to the reviewer’s comments is modified as the attached file.

Reviewer 2 Report

While the idea of the paper is interesting, the paper suffer from a lot of weakness and should be improved in many direction. Below are my comments:

The introduction should be improved, so that the motivation should be more clear and make the contribution very clear also. The authors should clearly identify the gap in the literature and show how the proposed approach can fill it.

It i is incorrect to say FDI log-returns. FDI is not an assets. Remove the word log-returns. use the word growth rate...

What you means by the "...the country cannot use the mean value of their..." at the end of the section of data analysis 3.1. Please correct this.

I cannot understand how the chart line of Singapore is the highest one and then in the table of data description of Table 1 it is ranked the third. Table 1 is it for FDI in level or in growth rate?

Results interpretation should be improved. Based on Table 3 you said that Indonesia is apparently 248 the nearest CE value (8.083) but it Table 1 it has a value of mean of 14. is there an explanation between the two?

Why you use the Vine-Copula if the analysis is bivariate?

Results interpretations and policy implications should be improved.

Finally, I recommend to the authors to improve the results interpretations from an economic point of view...

Additional references to be included if relevant:

L Charfeddine, N Benlagha (2016). A time-varying copula approach for modelling dependency: New evidence from commodity and stock markets. Journal of Multinational Financial Management 37, 168-189

Author Response

(The authors gave the same response as above.)

Round 2

Reviewer 1 Report

Thanks for improving the paper according to my points. 

Reviewer 2 Report

I am satisfied with the new version of the revised paper.

No new comments.